# Antihypertensive Potential of Japanese Quail (*Couturnix Couturnix Japonica*) Egg Yolk Oil (QEYO) in Sprague Dawley Rats

**DOI:** 10.3390/biology13040270

**Published:** 2024-04-18

**Authors:** Muhammad Sani Ismaila, Sherifat Olayemi Balogun-Raji, Fahad Hamza, Usman Bello Sadiya, Buhari Salisu, Mohammed Umar, Ishaka Aminu, Kegan Romelle Jones

**Affiliations:** 1Department of Basic Veterinary Sciences, School of Veterinary Medicine, Faculty of Medical Sciences, University of the West Indies, St. Augustine Campus, St. Augustine 999183, Trinidad and Tobago; 2Department of Veterinary Pharmacology & Toxicology, Faculty of Veterinary Medicine, Usmanu Danfodiyo University, Sokoto 840104, Nigeria; 3Department of Veterinary Surgery & Radiology, Faculty of Veterinary Medicine, Usmanu Danfodiyo University, Sokoto 840104, Nigeria; 4Department of Histopathology, Usmanu Danfodiyo University Teaching Hospital Sokoto, Sokoto 840104, Nigeria; 5Department of Medical Biochemistry, College of Health Sciences, Usmanu Danfodiyo University, Sokoto 840104, Nigeria

**Keywords:** acute and sub-chronic toxicity, BCOP test, HET-CAM irritation test, histological changes, hypertension, lipid profile, oxidative stress biomarkers, QEYO

## Abstract

**Simple Summary:**

Quail eggs have been reported as a nutraceutical in developed and developing countries. However, there are few documented reports on the effect of quail egg yolk oil on hypertension. In recent times, authors have identified anti-hypertensive compounds present in quail eggs. As such, this project was conducted to investigate the anti-hypertensive effect and toxicity levels of quail egg yolk oil. It was observed that quail egg yolk oil extracted using gentle heating and n-hexane reduced the blood pressure of Sprague Dawley rats. The toxicity studies showed no clinical or subclinical diseases at 28 days. This study demonstrated the anti-hypertensive effect of quail egg yolk oil, which can be used as an adjuvant therapy in eye drops or as a cosmetic agent.

**Abstract:**

Oils from animal sources have been used for centuries in the management of diseases. This research was conducted to screen the ex vivo and in vivo toxicity of quail egg yolk oil (QEYO) extracts and assess their effects on the management of hypertension in rats. QEYO was extracted using gentle heating (GH) and n-hexane (NHN). The extracts were subjected to toxicity testing using the hen’s egg test on chorioallantoic membrane (HET-CAM) and bovine corneal histology test. Acute and sub-chronic toxicity (28 days) were evaluated in rats. Hypertension was induced in rats by administering 80 mg/kg of N^ω^-L-Arginine Methyl Ester (L-NAME) per day for 28 days. Treatments commenced on the 14th day; Nifedipine at 30 mg/kg and 1 mL of distilled water were administered as positive and negative controls. Blood pressure (BP), lipid profiles, and oxidative stress markers were quantified. No irritation was observed using the HET-CAM test in the egg treated with both extracts. Bovine corneal histology showed no lesions in all treated groups. No signs of toxicity were observed in either acute or sub-chronic toxicity studies. A significant reduction in blood pressure was observed in rats treated with the extracts (*p* < 0.05). Changes in total cholesterol (TC), triglycerides (TGs), low-density lipoproteins (LDLPs), and high-density lipoproteins (HDLPs) were not significant compared to the control (*p* > 0.05). Oxidative stress markers (SOD and CAT) increased significantly in the treated groups compared to the control, while the malondialdehyde levels decreased (*p* < 0.05). QEYO was safe in both ex vivo and in vivo studies and can be said to have the potential to lower blood pressure as well as cardio-protective effects in hypertensive rats. This research provides evidence based on which QEYO could be used safely as an adjuvant therapy in eye drops and cosmetics and can be considered an effective choice for preventing hypertension.

## 1. Introduction

Egg yolk contains a major source of active principle usable in medical, pharmaceutical, cosmetic, nutritional, and biotechnological industries [1]. A recent survey carried out among farmers and buyers of quails and its products showed that even though there is no scientific backing concerning the use of quail egg and meat, all the people interviewed agreed that quail eggs and meat have medicinal value and are very effective against hypertension (100%) and diabetes (95.2%). Seventy-six percent of participants claimed the medicinal effects were more potent when the quail eggs were consumed raw. The majority (96.2%) of quail farmers reared these birds for their perceived medicinal properties [2]. In the past, recommendations were made to decrease egg consumption to limit the risk of cardiovascular diseases. However, recent research has shown that there was no correlation between egg consumption and an increase in plasma total cholesterol [3]. Quail eggs have the potential to be used in adjuvant therapies in many pharmaceutical preparations. It is gaining momentum in research on the management of various ailments, but there is a dearth of information regarding its toxicity. Thus, screening for toxicity of compounds intended for medicinal, pharmaceutical, or nutraceutical purposes is an important step in every biomedical research to ensure safety.

Egg yolk oil is also called ovum oil. It is generally derived from the yolk of chicken eggs, although it can be obtained from goose, duck, and other avian eggs [4]. Folkloric reports showed that chicken egg oil has many medicinal uses, and it has been used for its analgesic effects for several years [5]. Historically, the consumption of quail has been linked to curing tuberculosis, which led to the domestication and consumption of quail meat and eggs in the later part of the nineteen century in Japan [6]. Quail egg oils have also been used to heal surgical wounds in rabbits [7].

Egg yolk contains amino acids such as tryptophan and tyrosine [8]; bioactive compounds such as choline and gamma linoleic acid (GLA) are also present [9]. There are also small amounts of micronutrients such as vitamins A, D, and E [10]. Zeisel et al. [11] discovered that the proteins in egg yolk could act as potent inhibitors of human platelet aggregation. Chicken yolk oil has been observed to have anti-inflammatory, analgesic, and nociceptive properties [5], and another study showed that this oil has wound-healing properties in rats [12].

Hypertension is the persistent increase in blood pressure (BP) above 140/90 mmHg. Blood pressure in the arteries is persistently elevated. Blood pressure is expressed by two measurements, systolic and diastolic, which are the maximum and minimum pressures, respectively, in the arterial system [13]. Elevated blood pressure is a major risk factor for the development of cardiovascular disease, including stroke, myocardial infarction, renal diseases, and cardiac mortality [14]. Persistent elevation of blood pressure is an important public health issue globally because, even though it is readily detectable by routine blood pressure measuring, it can result in lethal complications if left untreated.

New drugs such as aprocitentan have been used to treat hypertension but possess adverse effects such as edema [15]. Lately, attention has been focused on herbs and minerals as potential therapeutic agents for the prevention and management of cardiovascular disease [16]. The objectives of this study were to assess the anti-hypertensive effect of QEYO extract whilst monitoring its toxicity using in vivo and in vitro techniques.

## 2. Materials and Methods

### 2.1. Ethical Statement

The research was carried out in the Department of Pharmacology and Toxicology, Usman Danfodiyo University Sokoto, while histological slides were processed and interpreted in the Department of Pathology, UDUTH, Sokoto Nigeria. The research was approved by the Faculty of Veterinary Research and ethics committee with approval number UDU/UREC/2021/036.

### 2.2. Drugs and Chemicals

All chemicals and drugs used in this experiment were of analytical grade. The following chemicals and drugs were obtained and used for this research: L-NAME and Nifedifine were purchased from Sigma-Aldrich, Berlin, Germany; and concentrated sulfuric acid (H_2_SO_4_) and ferric chloride were purchased from BDH Ltd., Poole, UK.

### 2.3. Quail Egg Material

The Japanese quail eggs were purchased from Kubwa Market, Abuja. The eggs were identified and authenticated by an Avian Pathologist from the Department of Veterinary Pathology, Usmanu Danfodiyo University, Sokoto (UDUS).

### 2.4. Preparation of Quail Egg Powder

A total of 240 quail egg yolks were manually separated from their egg whites [5], sundried for a day, and then blended until a powdery form was obtained. After blending, 800 g of quail egg yolk powder was obtained; this was divided into two equal parts for the extraction process of the egg yolk oil. Extraction of the quail egg yolk oil was performed using the *n*-hexane (QEYONH) extraction method and gentle heating (QEYOGH) method.

### 2.5. N-hexane Extraction of Quail Egg Yolk Oil

The oil content of Japanese quail egg yolk was extracted in the Pharmacology and Toxicology Department laboratory, Faculty of Pharmaceutical Science, UDUS, using a Soxhlet extractor (Illinois, Chicago, USA). Four-hundred grams of powdered egg sample was put into a porous thimble and placed in the Soxhlet extractor, using 150 cm^3^ of n-hexane (with a boiling point of 40–60 °C) as an extraction solvent for 6 h. The oil was obtained after the solvent was removed under reduced temperature and pressure and refluxing at 70 °C to remove the excess solvent from the extracted oil. The oil was then stored in freezer at −2 °C for subsequent studies [17,18].

### 2.6. Gentle Heating Extraction of Quail Egg Yolk Oil

The egg yolk oil was prepared from 400 g of egg powder. Yolk powder was gently heated for approximately 4 h and filtered using by a size 50 filter. The filtrate was used as the yolk oil prepared using direct heat [5,18].

### 2.7. HET-CAM Irritation Ex Vivo Toxicity Study

Luepke et al. [19] utilized embryonated 9-day-old chicken eggs and incubated them at 37.5 °C and a relative humidity of 63%. The eggs were candled to verify live embryos on the following day (their 10th day) in preparation for the HET CAM test. The procedure was performed under a well-illuminated airflow hood. The shell around the air cell of each egg was carefully removed, and the outer membrane was extracted using 30 cm blunt tweezers to reveal the chorioallantoic membrane after moistening with 250 µL of 0.9% NaCl (Figure 1). Approximately 200 µL of 0.1 M NaOH (positive control) was added to the membrane and left in contact for 20 s and then removed using tissue paper. The reaction was observed for 5 min, examining the membranes for hemorrhage, vascular damage (lysis), and/or coagulation, and the time taken for each to occur was noted and recorded. The same procedure as stated previously was repeated with 0.9% NaCl as the negative control. QEYO extracts obtained through gentle heating and n-hexane were administered, with three embryonated eggs per treatment [19].

### 2.8. Bovine Eye Histological Test

An in vivo toxicity test was carried out on the bovine eye, and toxicity was assessed based on damage to corneal tissue [20]. The toxicity was assessed based on damage to the epithelial, stroma, and endothelium.

Briefly, bovine eyes were collected soon after the death of the animal and placed in cool normal saline solution. The corneas were carefully examined to select those with no injury. Three corneas were assigned to each of the test substances (samples) and the control groups. The test substance, which was composed of NHN extract of QEYO, GH extract of QEYO, and 100% ethanol, served as a positive control, while normal saline served as a negative control. The corneas were then exposed to the test materials at a concentration of 0.1 mL and 0.2 mL of each test sample or control substance applied to the epithelial surface for 20 min (Figure 2). After exposure, the test article and controls were removed; the corneas were thoroughly rinsed at least 3 times with Eagle’s medium containing phenol and were incubated at 32 °C for 1 h. The corneas were then fixed in 75% alcohol. Histological changes in the corneal epithelium, stroma, and endothelium were graded as 1 (Slight), 2 (Mild), 3 (Moderate), or 4 (Severe).

### 2.9. Acute Toxicity Studies

Acute toxicity studies were adopted from Lorke [21] based on morbidity and mortality when these extracts were given to rats in feed and drinking water. Twenty-one matured rats of both sexes were randomly assigned into seven groups (1–7) of three rats per group. Groups 1–6 were administered QEYO from the two extracted samples at 1000 mg/kg, 3000 mg/kg, and 5000 mg/kg, respectively, via oral gavage. Group 7 rats received distilled water (10 mL/kg). The rats were allowed free access to feed and drinking water and were observed for 48 h for signs of toxicity and death.

#### 2.9.1. Sub-Chronic Toxicity Studies

For sub-chronic toxicity testing, twenty-one rats were used. They were divided into three groups of seven rats in each group. Group one was the control group, which received distilled water daily for twenty-eight (28) days. Rats in group 2 were orally administered QEYO extracted with n-hexane at 1000 mg/kg, while the rats in group 3 were given QEO extracted through gentle heating at 1000 mg/kg body weight for twenty-eight days. The standard acceptable toxicity limit is 1000 mg/kg. All the groups received feed and water ad libitum, with the following parameters to be analyzed.

#### 2.9.2. Hematological Parameters

The hematological parameters analyzed were the erythrocyte, thrombocyte, and total and differential leukocyte counts; hemoglobin concentration; hematocrit; mean corpuscular volume (MCV); mean corpuscular hemoglobin (MCH); and mean corpuscular hemoglobin concentration (MCHC).

#### 2.9.3. Serum Biochemistry

Blood samples in plain tubes were centrifuged at 3500 rpm to collect serum. The serum total protein, albumin, glucose, cholesterol, bilirubin, creatinine, urea concentration and alanine transaminase (ALT), aspartate transaminase (AST), γ-glutamyl transaminase and lactate dehydrogenase (LDH) activities were determined spectrophotometrically.

#### 2.9.4. Histopathological Examination

Slides of preserved organ samples were stained with hematoxylin and eosin [22]. The histological slides were examined, and the lesions were scored using the blind study method.

### 2.10. Animal Study for Antihypertensive Effect

Thirty Sprague Dawley rats of both sexes weighing between 113 and 193 g were housed at room temperature in an animal house in the Department of Pharmacology and Therapeutics, Usmanu Danfodiyo University Sokoto. They were all kept in standard cages and male and female animals within each group were demarcated from one another. The animals were fed on Vital Grower feeds^®^ (Vital Feeds, Jos, Plateau State, Nigeria), and water was provided ad libitum using plastic bottles.

#### 2.10.1. Induction of Hypertension and Dosing

Thirty Sprague Dawley rats were purchased from the Nigerian Institute of Trypanosomiasis Research (NITR), Kaduna Nigeria, and divided into 6 experimental groups containing 5 rats of either sex per group (*n* = 5). Hypertension was induced by administering L-NAME at a dose of 80 mg/kg/day in drinking water for 28 days. The dosing was administered via oral gavage to ensure animals consumed the appropriate amount of L-NAME in their drinking water. Quail egg yolk oil (QEYO) extracts were administered in 200 mg/kg and 300 mg/kg doses in the treatment groups in divided doses throughout the treatment period.

#### 2.10.2. Grouping

Group I animals were designated as a treatment group, and they received 200 mg/kg of quail egg yolk oil extracted using gentle heating (QEYOGH). Group II animals were dosed with 300 mg/kg of quail egg yolk oil extracted using gentle heating (QEYOGH). Group III animals were treated with 200 mg/kg of quail egg yolk oil extracted using *n*-hexane (QEYONH). Group IV animals received 300 mg/kg of quail egg yolk oil extracted using *n*-hexane (QEYONH). Group V animals were treated with nifedipine (NF) as a positive control at a dose of 30 mg/kg/day [23]. Group VI, designated as the control group (negative control), received 1 mL of distilled water per day.

Before the commencement of the study, all rats were allowed to acclimatize to the new environment for 2 weeks. Then, another 2 weeks were used to train the rats to remain calm in the blood pressure restrainers by placing each rat in the blood pressure restrainers for two minutes per day. Following this period of acclimatization, the base line reading (day 0) for systolic blood pressure (SBP) and diastolic blood pressure (DBP) for each animal was obtained. Treatment with extracts began from the 14th day after inducing hypertension in all groups with L-NAME (80 mg/kg/day) from day 0 to 14. The treatment lasted for 14 days in addition to the continuous administration on L-NAME (80 mg/kg/day) until the 28th day of the experiment.

#### 2.10.3. Screening for Antihypertensive Potential of Extracts

For the estimation of blood pressure, each rat was placed in the non-invasive blood pressure (NIBP) restrainer, and an appropriate cuff with a sensor was then mounted on its tail and warmed to approximately 33–35 °C. The tail cuff was inflated to a pressure well above the expected systolic blood pressure, i.e., 250 mmHg, and slowly released, during which the pulse was recorded using a UGO BASILE BP recorder, as were the systolic blood pressure (SBP) and diastolic blood pressure (DBP).

To ensure accurate BP determination, at least 4 consecutive readings of the maximum and minimum BP values were recorded. The average of the four maximum and minimum values was then calculated [24].

### 2.11. Blood Serum Collection and Lipid Profile Analysis

#### 2.11.1. Preparation of Serum

The procedure used for serum collection was as outlined in [25]. The rats were anesthetized with chloroform; when they became unconscious, a 10 mL syringe was inserted approximately 30 degrees from the horizontal axis of the sternum so that the needle entered the heart, and blood was drawn into sterile bottles [26]. The bottles were then left at room temperature for approximately 10 min, after which they were centrifuged at 33.5× *g* for 15 min. The sera obtained after centrifuging were then aspirated with a pipette into clean dry sterile bottles and frozen until required for analysis [27].

#### 2.11.2. Analysis of Serum Lipids

The serum total cholesterol concentration was estimated using the method described by Fredriskson et al. [28], while the serum low-density lipoprotein cholesterol (LDL-C) content was estimated as described by Dermacker et al. [29]. The method presented by Fossati and Prencipe [30] was adapted to determine the serum high-density lipoprotein cholesterol (HDL-C) concentration.

#### 2.11.3. Analysis of Oxidative Stress Markers

Blood serum previously collected and frozen from each group was tested for oxidative stress indicators such as malondialdehyde (MDA), superoxide dismutase (SOD), catalase (CAT), and glutathione peroxidase (GPx).

SOD was assayed using Cayman’s Superoxide Dismutase Assay Kit, Cayman Chemical, USA, following the manufacturer’s instructions. The assay utilizes tetrazolium salt for the detection of superoxide radicals generated by xanthine oxidase and hypoxanthine [31]. Measures of 200 µL of the diluted radical detector, 10 µL of prepared standard to the standard well, and 10 µL of serum to the sample well were added to each well, and 20 µL of diluted xanthine oxidase was added to both standard and sample wells to initiate the reaction. The plate was shaken for a few seconds to mix and was covered with a cover plate. The plate was then incubated on a shaker at room temperature for 20 min, and absorbance was read at 450 nm using a Rayto (RT 2100C) plate reader.

Catalase activity was assayed using Cayman’s Catalase Assay Kit following the manufacturer’s instructions. The method is based on the reaction of the enzyme with methanol in the presence of an optimal concentration of H_2_O_2_. The formaldehyde produced is measured spectrophotometrically with 4-amino-3-hydrazine-5-mercapto-1, 2, 4-triazole (Purpald) as the chromogen. Purpald specifically forms a bicyclic heterocycle with aldehydes, which, upon oxidation, changes from colorless to a purple-colored complex measured at 540 nm.

Three wells were designated in the micro titer plates as sample, standard, and control wells. To each well, 100 µL of assay buffer and 30 µL of methanol were added. A measure of 20 µL of prepared standard (formaldehyde standard) was added to the standard well, 20 µL of serum was added to the sample well, and 20 µL of H_2_O_2_ was added to each well to initiate the reaction. The plate was covered with a lid and incubated on a shaker for 20 min at room temperature. A measure of 30 µL of potassium hydroxide was added to each well to stop the reaction, and 30 µL of purpald was then added to produce a color change. The plate was covered once again and incubated for 10 min at room temperature on a shaker. Then, 10 µL of potassium periodate was added to each well, which were again covered and incubated for 5 min on a shaker. The absorbance of the mixtures was read at 540 nm using a Rayto (RT 2100C) plate reader.

#### 2.11.4. Glutathione Peroxidase Activity

The glutathione peroxidase activity was assayed using Cayman’s Glutathione Peroxidase Assay Kit, Cayman Chemical, Chicago, IL, USA, according to the manufacturer’s instructions. This assay measures glutathione peroxidase activity indirectly via a coupled reaction with glutathione reductase. Oxidized glutathione, produced upon the reduction of hydroperoxide by glutathione peroxidase, is recycled to its reduced state by glutathione reductase and NADPH:$$\text{R-O-O-H}+2\mathrm{GSH}\;\overset{Glutathion\;e\;Pexoxidase}{\to }\text{R-O-H}+\text{GSSG}+H_{2}O$$
$$\mathrm{GSSG}+\mathrm{NADPH}+H^{+}\;\overset{Glutathion\;e\;reductase}{\to }2\text{GSH}+{\text{NADP}}^{+}$$

The oxidation of NADPH to NADP^+^ is accompanied by a decrease in absorbance at 340 nm.

Three wells were designated as sample, non-enzymatic, and control wells. Measures of 100 µL of assay buffer, 50 µL of co-substrate mixture, and 20 µL of serum were added to the sample well; 120 µL of assay buffer and 50 µL of co-substrate mixture were added to the non-enzymatic well; and 100 µL of assay buffer, 50 µL of co-substrate mixture, and 20 µL of diluted GPx were added to the positive control well. The reaction was initiated by adding 20 µL of cumene hydroperoxide to each well, and the plate was carefully shaken for a few seconds to mix. The absorbance was read at 340 nm using a Rayto (RT 2100C) plate reader once every 3 min.
$$\Delta \mathrm{Abs}/\min=\frac{Abs(time2)-Abs(time1)}{Time2(\min)-Time1(\min)}$$
$$\mathrm{GPx}\;\mathrm{activity}=\frac{Abs/min}{0.00373\;µ/M}\;\times \;\frac{0.19\;mL}{0.02\;mL}=\mathrm{nmol}/\mathrm{mn}/\mathrm{mL}$$

#### 2.11.5. Estimation of Lipid Peroxidation

Lipid peroxidation (malondialdehyde) was estimated by measuring the amount of malondialdehyde, as evidenced by the formation of thiobarbituric acid reactive substances (TBARS), following the manufacturer’s instructions.

The assay is based on the reaction of malondialdehyde with thiobarbituric acid, forming an MDA-TBA_2_ adduct that absorbs strongly at 535 nm. Briefly, 0.1 mL of serum was added to a test tube and treated with 2 cm^3^ of (1:1:1 ratio) TBA–TCA–HCl reagent (0.37% thiobarbituric acid, 0.25N HCl, and 15% TCA). The tube was placed in a water bath for 15 min and cooled and centrifuged at room temperature for 10 min at 1000 rpm. The absorbance of clear supernatant was measured against a reference blank at 535 nm. The concentration of TBARS was calculated using the molar extinction coefficient of malondialdehyde (1.5 × 10^5^ mol/L/cm).

#### 2.11.6. Histopathological Analysis

At the end of the study, the animals were sacrificed via decapitation, and the hearts, kidneys, and livers were removed from the representative rats and stored in 10% buffered formalin for 48 h for histopathological analysis.

The livers, kidneys, and hearts were later impregnated with paraffin and made into tissue blocks. The paraffin-embedded tissues were sectioned with microtome and placed on labelled frosted end slides. After removing the paraffin with xylene, reverse dehydration took place in absolute to 70% alcohol, and staining was performed with hematoxylin and eosin stains. The slides were then examined under a light microscope, and photomicrographs of the section were taken [32].

#### 2.11.7. Statistical Analysis

Data obtained from this study have been presented in pictorial format, graphs, and tables for descriptive analysis. Repeated measures parametric analysis was used to identify the extraction method and concentration of quail egg yolk oil that lowered SBP and DBP the most from day 14 to 28, using In vivo Stat Statistical software version 3.0.0.0, 2008–2016, while MANOVA (multivariate analysis of variance) was used to compare the serum lipid levels among the extraction methods. Variables were compared for statistical significance at a probability of 5% (*p* > 0.05) with a confidence interval of 95%.

## 3. Results

### 3.1. HET-CAM Test

The HET-CAM irritation test showed that there was little to no irritation with the n-hexane and gently heated QEYO extract in comparison to the negative control. The positive control (0.2 mL Na OH) showed strong irritation, giving a value of 10.51%, as shown in Table 1 (see Figure 1B). Similarly, no irritation, coagulation of blood, or toxicity to the blood vessels after treatment were seen with QEYO extracts (see Figure 1C,D).

### 3.2. Bovine Corneal Gross Appearance

The bovine eyes inoculated with the two extracts of QEYO at two different concentrations showed no gross lesions on the cornea. However, the positive control group treated with 0.1 mL of 100% ethanol does show trauma to the cornea (Figure 2).

### 3.3. Histology of the Bovine Cornea

Figure 3 shows the histological sections of the bovine cornea of the positive and negative controls as well as the QEYO extract treatment groups. The negative control non-treated group shows an intact thick stratified squamous epithelial section compared to the positive control, which was thin and destroyed (see Figure 3(1a,1b)). Similarly, the stratified squamous epithelial lining of the bovine cornea inoculated with 0.1 mL QEYONH and 0.1 mL and 0.2 mL of QEYOGH extract as shown in Figure 3(2a,3a,4a), respectively. In Figure 3, the collagenous stroma is intact in the negative control group as well as the QEYOGH group. The positive control group shows partial destruction of the stroma indicated in blue thick arrow (1b). In Figure 3(1a,3b,4a) the Descemet’s membrane is intact. Which is prominent in the negative control group, QEYONH and QEYOGH groups at various concentrations compared to the positive control.

### 3.4. Acute In Vivo Toxicity Study

For the acute toxicity study, after the oral administration of QEYO of the two different extracts at three different doses (1000 mg/kg, 3000 mg/kg, and 5000 mg/kg), rats showed no sign of toxicity after 48 h of monitoring. In a sub-chronic toxicity study, the two extracts of the QEYO were administered at the same dose rate of 1000 mg/kg for a period of 28 days orally with the control group receiving distilled water.

### 3.5. Animal Weight and Behavioral Changes

All rats treated with quail egg yolk oil (QEYO) extract from the two methods showed normal feed and water intake. The treatment did not cause any behavioral changes or mortality in the rats. In addition, there was no sign of illness in the rats throughout the course of treatment. An increase in body weight was observed in the treated rats and the negative control during the four-week period (Table 2).

### 3.6. Hematological Parameters

No significant changes in the hematological parameters were observed after 28 days of treatment with quail egg yolk oil (QEYO) extracts compared to the control non-treated group (*p* > 0.05). However, there were numerical increases in the MCV and quantities of neutrophils, monocytes, and lymphocytes in the group treated with the n-hexane fraction of the extract (Table 3).

### 3.7. Serum Biochemistry

The serum biochemistry results showed that there were no significant differences amongst the parameters measured in the various treatment of rats exposed to QEYO extracts (Table 4).

### 3.8. Histopathological Changes

The liver, kidney, and brain did not show any histological abnormality under light microscopy in treatment or control groups (Figure 4).

### 3.9. Blood Pressure

#### 3.9.1. Mean Baseline Systolic and Diastolic Blood Pressure of Rats at Day 0

The mean baseline reading at day 0 for systolic blood pressure obtained ranged from 123.2 ± 2.8 to 126.0 ± 2.7 S.E.M, while diastolic blood pressure ranged from 84.2 ± 1.8 to 86.6 ± 1.8 S.E.M. There was no significant difference (*p* > 0.05) between all the treated and control groups for the diastolic and systolic blood pressure at day 0.

#### 3.9.2. Effect of L-NAME on SBP and DBP of Rats within Groups after 14 Days of Administration

After the administration of L-NAME at 80 mg/kg/day for 14 days, systolic and diastolic blood pressure were similar in in the QEYOGH 200 and 300 mg, QEYONH 200 and 300 mg, positive control (nifedipine), and negative control groups, respectively (see Table 5 and Table 6).

#### 3.9.3. Comparison of SBP in All Treatment Groups from Day 0 to 28

There was a progressive rise in all blood pressure in all the rats after the administration of L-NAME (80 mg/kg/day). At day 21 of treatment, there was decrease in systolic blood pressure at the various doses of extracts. The changes in blood pressure reading at day 0 can be seen to move upwards along the plot as L-NAME is administered at 80 mg/kg/day in all treatment groups. At day 21 of treatment, there was a decrease in all groups except the control (negative) group, which increased steadily along the slope.

There was significant reduction in blood pressure in rats treated with QEYOGH at 300 mg/kg, Nifedipine (positive control), and QEYOGH at 200 mg/kg (*p* < 0.05) (Figure 5).

#### 3.9.4. Effect of L-NAME and QEYO on DBP Measurements in All Treatment Groups from Day 0 to 28

The DBP increased steadily in all treatment groups from day 7 to 14 after L-NAME administration, which is significant in all the groups. However, due to the effect of QEYO in different concentrations and extraction methods, DBP began to decrease from day 14 to 28. This reduction was significant in all groups except the QEYONH 200 mg/kg group. The control group had an elevated DBP reading, which was significant at *p* < 0.0001 (Figure 6).

### 3.10. Effect of QEYO on the Lipid Profile of L-NAME Hypertension Rats after 14 Days of QEYO Administration

Table 7 shows the effects of the administration of QEYOGH, QEYONH and Nifedipine groups on the serum lipid profile of L-NAME hypertension-induced rats compared to the negative control group.

The total cholesterol (TC) level increased in the QEYOGH and QEYONH groups; this increase was, however, not statistically significant (*p* > 0.05). There was no significant difference in TC levels within the QEYOGH group (QEYOGH at 200 mg/kg and QEYOGH at 300 mg/kg).

Triglyceride levels were also elevated in the treatment group compared to the control group; this elevation was, however, not significant (*p* > 0.05). Within the treatment groups, the increased dose in the QEYOGH and QEYONH groups did not also result in a significant difference (*p* > 0.05).

High-density lipoprotein levels were elevated in all treatment groups compared to the control group. However, this increase was not significant (*p* > 0.05). Rats treated with QEYOGH at 200 and 300 mg/kg showed the highest levels for HDLP.

Low-density lipoprotein levels were reduced in the rats that received QEYOGH and QEYONH at 200 mg/kg compared to the control group; this was, however, not significant, and all other treatment groups had elevated LDLP levels compared to the control, which was also not significant at *p* > 0.05 (Table 7).

### 3.11. Biomarkers of Oxidative Stress in Rats Treated with QEYOGH and QEYONH

Interesting results were obtained from the biomarkers of oxidative stress in rats fed QEYO extract (n-hexane and gentle heating). These extracts increased CAT, GP_x_, and SOD and reduced MDA in comparison with the negative control (*p* < 0.05). However, in comparison to the positive control, the QEYO extract caused lower levels of CAT, GP_x_, and SOD and higher levels of MDA. Amongst the various concentrations of QEYO extract, gentle heating at 300 mg/kg gave the best results as an antioxidant (Table 8).

### 3.12. Histopathological Findings in Tissues of Rats Treated with QEYOGH and QEYONH

There were no observable lesions on the histopathology of the heart in rats treated with QEYO extracts at 300 mg/kg administered for 14 days with concurrent administration of L-NAME at 80 mg/kg/day. There was a slight enlargement of the myocytes and coronary artery of the heart in rat induced with hypertension using L-NAME (80 mg/kg/day) and treated with QEYONH at 300 mg/kg as shown in Figure 7 Plate 3.

## 4. Discussion

In drug discovery, toxicity studies play a very critical role. Screening for toxicity in any substance that is intended for medicinal, cosmeceutical, or nutraceutical use is an important factor that can never be neglected. In vitro toxicity testing serves as a fundamental stage for testing the toxicity of compounds intended for medicinal or cosmeceutical purposes. In this research, both the HET-CAM and bovine histology evaluation of the toxicity level were used due to their rapid and accurate significance for measuring toxicity levels. The BCOP test, which assesses opacity and permeability endpoints, can be used to determine the damage to the corneal tissue caused by chemicals. However, it is challenging to identify chemicals that react with cellular targets such as nucleic acids or mitochondrial proteins without causing immediate loss of cellular integrity or protein precipitation. Histological evaluation of treated corneas is necessary for the identification of such chemicals [33,34]. Thus, evaluating the corneal tissue histologically provides a direct measurement of injury depth [35].

Only two out of four types of in vitro irritation toxicity tests were used in this study: HET-CAM and bovine corneal histology test [36,37]. Teixeira [38] reported that the method of toxicity assessment used for a substance is appropriate for general assessment and can be a useful tool to achieve the objectives of the 3Rs program (Refine, Reduce, and Replacement). Both BCOP and HET-CAM methods are valid and can be reliably applied in research on potential toxic substances. Our study found that QEYO did not show any signs of toxicity or irritation on the CAM, and the histology of the bovine eye revealed no irritation after treatment with the two different extracts of QEYO. L-NAME-induced hypertension is a method of inducing hypertension in animal research models. It is characterized by a generalized deficiency of NO and a progressive increase in BP when prolonged [39]. The L-NAME model is a good representation of hypertension in humans, making it ideal for studying the cardiovascular effects of new treatments. Prior research on quail egg yolk oil (QEYO) found that it contains sodium (Na), potassium (K), calcium (Ca), magnesium (Mg), and phosphorus (P). The concentration of potassium, sodium, and phosphorus was slightly higher than that of calcium and magnesium when QEYO was obtained via gentle heating as compared to n-hexane extraction [40]. Potassium supplementation has been proven to reduce blood pressure and mitigate complications in individuals with hypertension [41]. Thus, QEYO could be recommended as an anti-hypertensive agent. Several studies have shown that a reduced sodium intake is effective in lowering blood pressure and protecting the heart against various cardiovascular diseases. This may explain why QEYO has cardio-protective properties due to its low sodium concentration. Magnesium and, more recently, calcium levels have been found to have significant cardio-protective effects on the heart and lower blood pressure [42].

The production of nitric oxide (NO) by endothelial cells is responsible for the vasodilatory effect that regulates blood pressure. This process is dependent on factors that increase intracellular calcium concentration [43]. Previous research has shown that quail egg yolk oil extracted using gentle heating has a higher concentration of calcium (0.90 mg/L), which could be the reason for the observed decrease in blood pressure. A high calcium diet has plausible antihypertensive mechanisms, including decreased α1-adrenoceptor responsiveness [44,45], improved function of the cell membrane Na^+^-K^+^-ATPase, and reduced voltage-dependent calcium entry in arterial smooth muscle [46].

Ca supplementation may also augment arterial sensitivity to nitric oxide (NO) and enhance hyperpolarization of vascular smooth muscle [47], which may be one of the reasons for the lowered blood pressure observed in the QEYO treatment groups.

Saponins were reported to be obtained in significant amount in QEYO [40]. Saponins aid in reducing cholesterol levels by forming complexes with cholesterol and bile acids, which prevent them from being absorbed through the small intestine, thus lowering the cholesterol level in the blood and liver [48]. Saponins have been shown to be useful in maintaining high-density lipoprotein cholesterol (HDL-C) levels and lowering low-density lipoprotein cholesterol (LDL-C) levels, as reported earlier [49,50].

This study has shown that QEYO has the potential to lower blood pressure. This effect may be due to the high levels of antioxidants found in QEYO [51]. Previous research has shown that these antioxidants can improve the activity of antioxidant enzymes and up-regulate antioxidant genes in fruit flies [51]. This study further confirmed the positive effect of QEYO on antioxidants, as the treated group showed a significant increase in antioxidant biomarkers compared to the control group. These biomarkers help to boost nitrous oxide production [52], which contributes to the regulation of blood flow and pressure from the non-adrenergic, non-cholinergic terminals [53,54]. The mean decrease in SBP for all QEYOGH-treated groups was compared to the mean decrease in all SBP QEYONH groups, and there was no statistically significant difference (*p* > 0.05) between the groups. This shows that QEYO extracted through either the gentle heating method or n-hexane method could reduce blood pressure. These findings partially contradict the findings that quail eggs, either raw, fried, or boiled, consumed by anemic female hypertensive patients did not reduce blood pressure but rather increased the lipid profile in blood serum in humans [54].

High cholesterol levels in the blood are linked to cardiovascular diseases such as atherosclerosis, coronary heart diseases, and hypertension. The level to which elevated blood lipids contribute to these heart conditions is dependent on the distribution of the various types of lipoprotein classes in blood serum [55]. High levels of lipids except for high-density lipoproteins are associated with hypertension and atherosclerosis, while high levels of triglycerides and LDLP are associated with coronary artery diseases [55].

The significant levels of high-density lipoprotein (HDLP) in the treated rats show that QEYO has some cardio-protective abilities as it has been shown that high levels of HDLP in blood serum may exact some protective effects against atherosclerosis and hypertension and promote the mobilization and metabolism of cholesterol, thus reducing the deposition of cholesterol in the arterial blood vessels. Similarly, HDLP has also been shown in vitro to have some form of mechanism for the removal of peripheral cholesterol as well as very low-density lipoprotein (VLDL) and LDLP [55].

In this study, HDLP levels were observed to be significantly (*p* < 0.05) elevated in all the treated groups compared to the control group that did not receive QEYO, while the TC, TG, and LDLP levels were elevated compared to the control groups, but these increases were not significant (*p* < 0.05). This shows that QEYO does not totally increase the risk of cardiovascular diseases and hypertension, as opposed to early reports [52]. This study also reveals that QEYO has some anticholesteremic and antilipidemic effects. Several epidemiological studies have linked low intake of dietary antioxidants to an increased frequency of hypertension; this is in addition to the inverse relationship between heart disease and plasma antioxidants levels [54]. Endothelial cells of the vascular smooth muscles are particularly susceptible to oxidative stress, not only through ROS-mediated cell death but also because of the bioavailability of the normally protective mediator such as nitric oxide. Our previous study [51] reveals that QEYO contains oleic acid and diltiazem. Oleic acid has been shown to decrease the myocardial infarction rate, platelet aggregation, and secretion of TXA2, reduce systolic blood pressure, and improve immunity [56]. On the other hand, diltiazem is a calcium channel blocker that is clinically used as an antihypertensive, anti-arrhythmic, and anti-anginal agent for the management of cardiovascular conditions such as hypertension, chronic stable angina, atrial fibrillation, and atrial flutter [57]. Studies have revealed that egg yolk contains proline and arginine, both of which may decrease blood pressure, which may be through the inhibition of angiotensin-converting enzyme [3,58,59].

## 5. Conclusions

Based on the results obtained in this study, it can be concluded that quail egg yolk oil (QEYO) extracted using gentle heating and *n*-hexane has no toxic effects. These extracts also lower blood pressure and have some cardio-protective potential, as observed in the pathology of the heart in hypertension induced by L-NAME in rats. QEYO was efficient in improving heart functions and reducing the dyslipidemia effect of L-NAME-induced hypertension.

## Figures and Tables

**Figure 1 biology-13-00270-f001:**
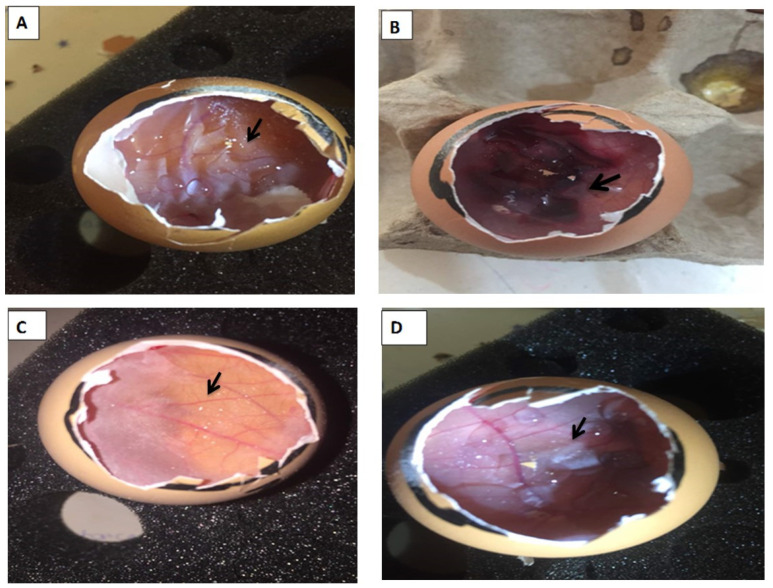
HET-CAM irritation. The arrows indicate observation areas on the egg CAMs. (**A**) Before treatment with 0.1 mL Na OH (negative control showing no sign of toxicity or coagulation on the blood vessels); (**B**) after treatment with 0.2 mL Na OH (positive control showing irritation and coagulation of blood on the vessels); (**C**) after treatment with 0.2 mL n-hexane quail egg yolk oil extract (showing no irritation or coagulation of blood); (**D**) after treatment with 0.2 mL gentle heating quail egg yolk oil (showing no toxicity to the blood vessels).

**Figure 2 biology-13-00270-f002:**
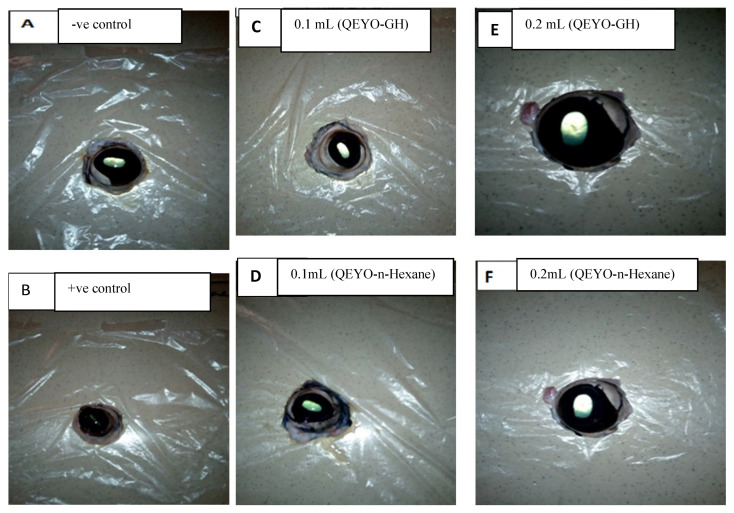
Gross appearance of the bovine cornea after inoculation with the positive control and different concentrations of the test substance: (**A**) 0.1 mL normal saline (negative control); (**B**) 100% ethanol (positive control); (**C**) 0.1 mL (QEYO-GH); (**D**) 0.1 mL (QEYO-n-Hexane); (**E**) 0.2 mL (QEYO-GH); (**F**) 0.2 mL (QEYO-n-Hexane).

**Figure 3 biology-13-00270-f003:**
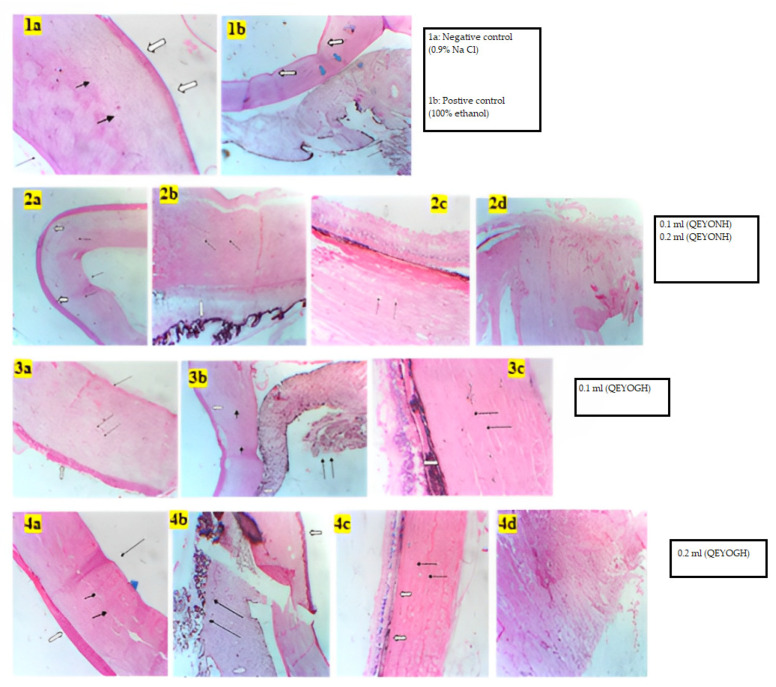
Histopathological sections of bovine corneas showing the negative control (0.1 mL of 0.9%NaCl) and positive control (0.1 mL of 100% ethanol) as well as the QEYONH (0.1 and 0.2 mL)-treated and QEYOGH (0.1 and 0.2 mL)-treated groups. (**1a**) Corneal stratified squamous epithelium (white arrow), collagenous stroma (black arrows), and Descemet’s membrane (long black arrow); (**1b**) corneal stratified squamous epithelium (white arrow), collagenous stroma (blue arrows), and ciliary body (double black arrows). (**2a**) Corneal stratified squamous epithelium (white arrow), collagenous stroma (black arrows), and Descemet’s membrane (long black arrow); (**2b**) ciliary body (white arrows) and scleral stroma (double black arrows); (**2c**) retina (white arrow) and scleral stroma (double black arrows); (**2d**) optic nerve. (**3a**) Corneal stratified squamous epithelium (white arrow), collagenous stroma (double black arrows), and Descemet’s membrane (long arrow); (**3b**) corneal stratified squamous epithelium (white arrow), collagenous stroma (black arrows head), and ciliary body (double black arrows); (**3c**) retina (white arrow) and scleral stroma (double black arrows). (**4a**) Corneal stratified squamous epithelium (white arrow), collagenous stroma (double black arrows), and Descemet’s membrane (long arrow); (**4b**) corneal stratified squamous epithelium (white arrow), collagenous stroma (black arrows head), and ciliary body (double black arrows); (**4c**) retina (white arrow) and scleral stroma (double black arrows); (**4d**) the optic nerve. All the slides were stained using hematoxylin and eosin and viewed using light microscopy at a ×100 magnification.

**Figure 4 biology-13-00270-f004:**
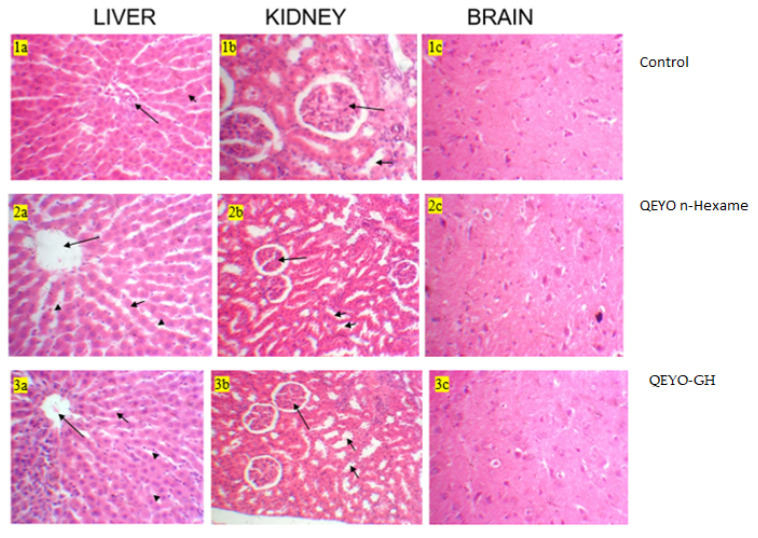
Histological sections of the liver, kidney, and brain. (**1a**) Liver: section show normal portal triad (arrow) and hepatocytes arranged in cords (short arrow); H&E × 100. (**1b**) Kidney: section shows regular glomerulus (Long arrow) and renal tubules (short arrow); H&E × 100. (**1c**) Brain: section shows regular neutrophils; H&E × 100. (**2a**) Liver: section shows normal hepatic central vein (long arrow), hepatocytes arranged in cords (short arrow), and sinusoidal space (arrow head); H&E × 100. (**2b**) Kidney: section shows regular glomerulus (long arrow) and renal tubules (short arrow); H&E × 100. (**2c**) Brain: section show regular neutrophils; H&E × 100. (**3a**) Liver: section show normal hepatic central vein (long arrow), hepatocytes arranged in cords (short arrow), and sinusoidal space (arrow head); H&E × 100. (**3b**) Kidney: section shows regular glomerulus (long arrow) and renal tubules (short arrow); H&E × 100. (**3c**) Brain: section show regular neutrophils; H&E × 100.

**Figure 5 biology-13-00270-f005:**
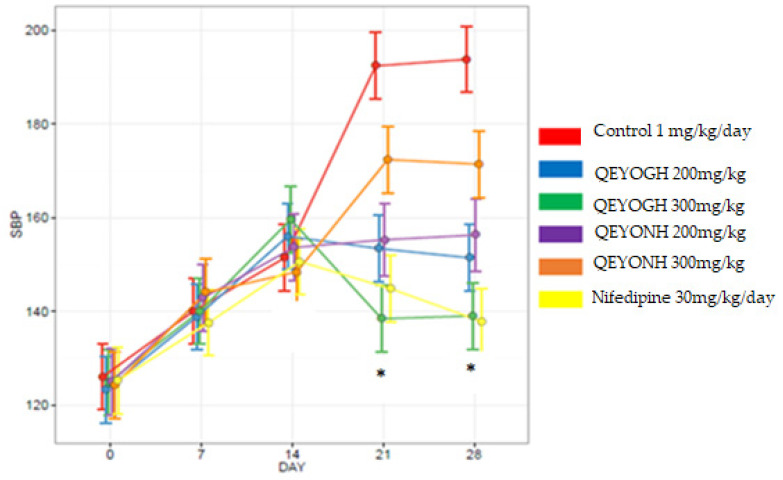
The mean SBP plot comparison from day 0 to day 28. Systolic blood pressure of rats treated with L-NAME (80 mg/kg) followed by treatment with different doses of quail egg yolk oil extracted using gentle heating (QEYOGH) and quail egg yolk oil extracted using *n*-hexane (QEYONH). Nifedipine (30 mg/kg/day) was used as a standard control drug. * Indicates a significant decrease in SBP in the groups treated with the higher doses of the extract and the positive control compared to the control non-treated group at 21 and 28 days (*p* < 0.05).

**Figure 6 biology-13-00270-f006:**
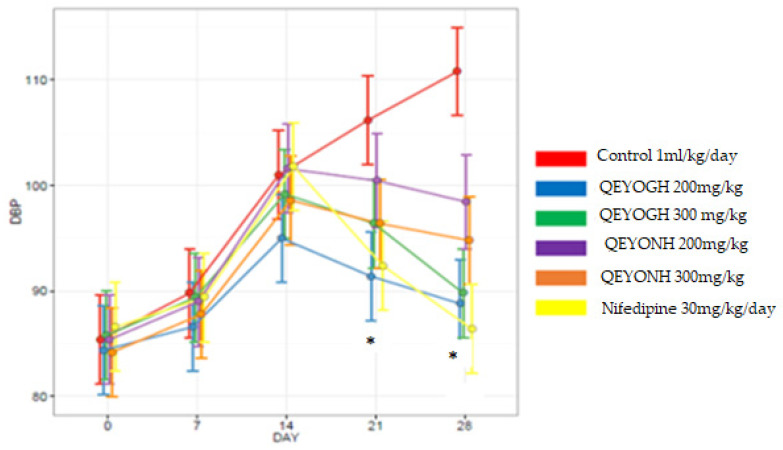
Diastolic blood pressure of rats treated with L-NAME (80 mg/kg) followed by treatment with different doses of quail egg yolk oil extracted using gentle heating (QEYOGH) and quail egg yolk oil extracted using *n*-hexane (QEYONH). Nifedipine (30 mg/kg/day) was used as a standard control drug. * Indicates a significant decrease in DBP in the groups treated with the higher doses of the extract and the positive control compared to the control non-treated group at 21 and 28 days (*p* < 0.05).

**Figure 7 biology-13-00270-f007:**
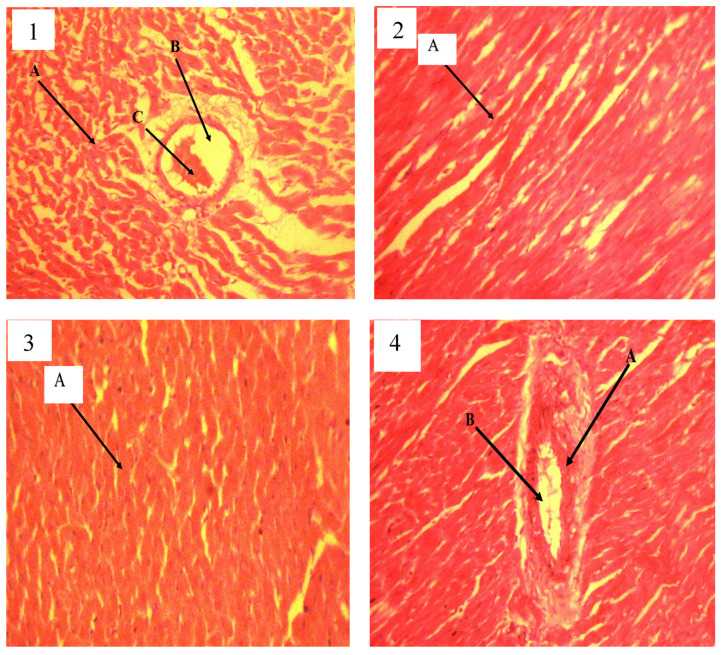
**Plate 1**: Photomicrograph of the heart of a rat induced with hypertension using L-NAME (80 mg/kg/day) and treated with 1 mL/kg of distilled water showing loss of architecture of the heart and oedema. There is also narrowing of the coronary artery due to thickness around the arterial lumen, H&E × 100; A = loss of architecture, B = narrowing of arterial lumen, C = arteriosclerosis. **Plate 2**: Photomicrograph of the heart of a rat induced with hypertension using L-NAME (80 mg/kg/day) and treated with nifedipine at 30 mg/kg/day showing normal architecture of the heart with normal myocytes and coronary artery of the heart, H&E × 100; A = normal heart wall. **Plate 3**: Photomicrograph of the heart of a rat induced with hypertension using L-NAME (80 mg/kg/day) and administered QEONH at 300 mg/kg showing slight enlargement of the myocytes and coronary artery of the heart, H&E × 100; A = slight enlargement of arterial wall. **Plate 4**: Photomicrograph of the heart of a rat induced with hypertension using L-NAME (80 mg/kg/day) and treated with quail egg yolk oil obtained by gentle hearting at 300 mg/kg showing normal architecture of the heart with normal myocytes and coronary artery, H&E × 100; A = coronary arterial wall, B = arterial lumen.

**Table 1 biology-13-00270-t001:** HET-CAM irritation score test for the quail egg yolk oil extracted using n-hexane and gentle heating.

Reagents	HET-CAM (Average)	Score Assessment
0.1 mL Na OH	10.51	Strong irritation
0.9% Na Cl	0	Nonirritant
n-hexane quail egg yolk oil extract	0.092	Practically no irritation
Gentle heating quail egg yolk oil extract	0.087	Practically no irritation

**Table 2 biology-13-00270-t002:** Mean ± SD of relative weight of the rats treated with QEYO extracted using n-hexane (NH) and gentle heating (GH) for 28 days.

Time/Group	Control	NH (1000 mg/kg)	GH (1000 mg/kg)
Week one	97.80 ± 6.18	127.60 ± 11.97	132.60 ± 12.01
Week two	107.20 ± 8.44	138.60 ± 12.72	143.80 ± 13.03
Week three	116.40 ± 11.37	152.60 ± 13.72	154.40 ± 13.13
Week four	131.20 ± 11.39	167.20 ± 11.71	163.40 ± 10.67

No significant differences were noted between the control and treated groups (*p* > 0.05).

**Table 3 biology-13-00270-t003:** Mean ± SD of relative complete blood count of the rats treated with QEYO extracted using n-hexane and gentle heating for 28 days.

Parameters	Control	N-Hexane Group (1000 mg/kg)	Gentle Heating Group (1000 mg/kg)
PCV (%)	35 ± 2	36 ± 2	36 ± 3
RBC	6.76 ± 0.95	6.37 ± 0.58	6.73 ± 1.07
Hb (g/dl)	11.42 ± 0.95	11.64 ± 0.79	11.70 ± 1.02
MCV (fL)	52.55 ± 4.53	56.30 ± 2.24	53.73 ± 4.36
MCH (pg/cell)	17.20 ± 1.44	18.20 ± 1.15	17.29 ± 1.42
MCHC (g/dL)	32.42 ± 1.39	32.51 ± 0.47	32.69 ± 0.77
WBC (109/L)	7.01 ± 2.03	6.97 ± 3.26	7.05 ± 2.01
Neutrophils (%)	11.00 ± 4.18	15.6 ± 4.67	12.6 ± 4.827
Lymphocyte (%)	84.60 ± 4.45	77.8 ± 5.26	80.20 ± 5.22
Monocytes (%)	5.60 ± 4.23	6.2 ± 2.28	5.60 ± 3.65
Band	1.33 ± 0.58	1 ± 1	2 ± 1.41

WBC = white blood count, RBC = red blood count, Hb = hemoglobin; PCV = pack cell volume, MCH = mean cell hemoglobin, MCV = mean cell volume; MCHC = mean cell hemoglobin concentration. No significant differences were noted between the control and treated groups (*p* > 0.05).

**Table 4 biology-13-00270-t004:** Mean ± SD of relative serum biochemistry of the rats treated with QEYO extracts using n-hexane and gentle heating for 28 days.

Parameters	Control	NH (1000 mg/kg)	GH (1000 mg/kg)
Albumin (g/dl)	2.74 ± 0.47	2.92 ± 0.47	2.84 ± 0.40
T. protein (g/dl)	6.26 ± 0.24	6.40 ± 0.37	6.32 ± 0.34
D. bilirubin (um)	0.31 ± 0.06	0.53 ± 0.30	0.42 ± 0.31
T. bilirubin (um)	1.68 ± 0.30	1.77 ± 0.37	1.64 ± 0.43
Urea (mmol/L)	5.80 ± 0.16	4.40 ± 0.46	4.52 ± 1.13
Creatinine (mmol/L)	0.50 ± 0.10	0.70 ± 0.29	0.62 ± 0.18
Cholesterol (mmol/L)	90.00 ± 4.86	95.80 ± 7.82	85.00 ± 4.24
ALT (IU/L)	565.80 ± 1.11	598.20 ± 1.82	535.00 ± 2.14
AST (IU/L)	255.60 ± 1.14	270.60 ± 4.51	221.40 ± 6.19
LDH (IU/L)	2.20 ± 0.84	2.40 ± 1.14	2.80 ± 1.64

AST = aspartate aminotransferase, ALT = alanine aminotransferase LDH = lactate dehydrogenase. No significant differences were noted between the control and treated groups (*p* > 0.05).

**Table 5 biology-13-00270-t005:** Effect of L-NAME on SBP and DBP of rats within groups after 14 days of administration.

Treatment	Mean Difference of BP Reading	
	SBP	DBP
QEYOGH (200 mg/kg)	32.8 ± 4.3	10.6 ± 1.8
QEYOGH (300 mg/kg)	34.8 ± 4.3	13.4 ± 1.8
QEYONH (200 mg/kg)	28.6 ± 4.3	16.2 ± 1.8
QEYONH (300 mg/kg)	24.2 ± 4.3	14.4 ± 1.8
Positive control (Nifedipine 30 mg/kg/day)	25.4 ± 4.3	15.4 ± 1.8
Negative Control (1 mL distilled water/day)	25.4 ± 4.3	15.6 ± 1.8

**BP** = blood pressure **SBP** = systolic blood pressure **DBP** = diastolic blood pressure **QEYOGH** = quail egg yolk oil gentle heating **QEYONH** = quail egg yolk oil *n*-hexane. No significant differences were noted between the control and the treated groups (*p* > 0.05).

**Table 6 biology-13-00270-t006:** Differences in blood pressure values (SBP and DPB) in L-NAME hypertension-induced rats within treatment groups between days 14 and 28.

Treatment	Mean Difference of BP Reading	
	SBP	DBP
QEYOGH (200 mg/kg)	−4.6 ± 4.3 ^b^	−6.2 ± 1.8 ^b^
QEYOGH (300 mg/kg)	−20.6 ± 4.3 ^a^	−9.4 ± 1.8 ^b^
QEYONH (200 mg/kg)	−2.6 ± 4.7 ^b^	−3.1 ± 2.0 ^a^
QEYONH (300 mg/kg)	−23.0 ± 4.3 ^a^	−3.8 ± 1.8 ^a^
Positive control (Nifedipine 30 mg/kg/day)	−12.8 ± 4.3 ^c^	−15.4 ± 1.8 ^c^
Negative control (1 mL distilled water/day)	42.4 ± 4.3 ^d^	9.8 ± 1.8 ^d^

**BP** = blood pressure **SBP** = systolic blood pressure **DBP** = diastolic blood pressure **QEOGH** = quail egg yolk oil gentle heating **QEYONH** = quail egg yolk oil *n*-hexane. ^a,b,c,d^ Superscripts denote significant differences (*p* < 0.05) within columns.

**Table 7 biology-13-00270-t007:** Effect of QEYO administration on serum lipid profile of L-NAME hypertensive rats.

Treatment	TC	TG	HDLP	LDLP
QEYOGH (200 mg/kg)	1.28 ± 0.10	0.63 ± 0.12	0.84 ± 0.45	0.10 ± 0.03
QEYOGH (300 mg/kg)	1.78 ± 0.89	0.89 ± 0.36	1.04 ± 0.02	0.32 ± 0.20
QEYONH (200 mg/kg)	1.34 ± 0.17	0.97 ± 0.24	0.58 ± 0.07	0.16 ± 0.07
QEYONH (300 mg/kg)	1.49 ± 0.34	0.79 ± 0.25	0.58 ± 0.22	0.14 ± 0.15
Positive control (Nifedipine 30 mg/kg/day)	1.39 ± 0.00	0.81 ± 0.31	0.54 ± 0.18	0.38 ± 0.31
Negative Control (1 mL Distilled water)	1.22 ± 0.40	0.49 ± 0.03	0.49 ± 0.32	0.17 ± 0.11

**TC** = total cholesterol **TG** = total triglyceride **HDLP** = high-density lipoproprotein **LDLP** = low-density lipoprotein **QEYOGH** = quail egg yolk oil gentle heating **QEYONH** = quail egg yolk oil *n*-hexane. No statistical significance was recorded in lipid profile when comparing the values between the control and the treatment groups (*p* > 0.05).

**Table 8 biology-13-00270-t008:** The effect of QEYOGH and QEYOGH on biomarkers of oxidative stress in rats.

Treatment	CAT	GPx	SOD	MDA
QEYOGH at 200 mg/kg	27.0 ± 0.8 ^c^	29.7 ± 1.5 ^b^	28.6 ± 0.5 ^d^	0.44 ^d^
QEYOGH at 300 mg/kg	35.1 ± 0.8 ^d^	69.6 ± 2.9 ^a^	48.6 ± 2.1 ^e^	0.33 ^c^
QEYONH at 200 mg/kg	15.3 ± 0.1 ^b^	26.3 ± 1.5 ^b^	23.2 ± 0.5 ^c^	0.52 ^b^
QEYONH at 300 mg/kg	15.7 ± 00 ^b^	31.4 ± 1.5 ^b^	17.7 ± 0.3 ^b^	0.52 ^b^
Positive control (Nifedipine 30 mg/kg/day)	33.8 ± 0.1 ^d^	58.6 ± 6.7 ^a^	59.3 ± 1.8 ^e^	0.28 ^e^
Negative Control (1 mL Distilled water)	10.9 ± 0.4 ^a^	10.2 ± 2.5 ^a^	8.6 ± 1.1 ^a^	0.84 ^a^

**CAT** = catalase, **GPx** = glutathione peroxidase, **SOD** = superoxide dismutase, **MDA** = malondialdehyde, **QEYOGH** = quail egg yolk oil gentle heating, **QEYONH** = quail egg yolk oil *n*-hexane. ^a–e^ Superscripts denote significant differences (*p* < 0.05) within columns.

## Data Availability

The data are available upon request.

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
