# Peer review of "Antihypertensive Potential of Japanese Quail (Couturnix Couturnix Japonica) Egg Yolk Oil (QEYO) in Sprague Dawley Rats"

_biology, 2024, doi:10.3390/biology13040270_

Round 1

Reviewer 1 Report

Comments and Suggestions for Authors

This is a very interesting and potentially important study examining the toxicity of quail egg oil. Since quail egg oil may have a large number of beneficial properties, making it suitable for a variety of uses, identifying any potential toxic side effects greatly enhances understanding of when it is appropriate to use it medically and in other commercial applications. I like and appreciate the historical and ethnographic perspective in the introduction, explaining traditional and current practices using egg oils. Combining this with the extensive scientific testing described here is a great strategy for developing new approaches to maintaining and improving health.

Comments on the Quality of English Language

I commend the authors for writing a manuscript so well in a language that is not their native one. I would not do nearly this well if trying to write in a language other than English. However, there are several places where phrasing is awkward… not necessarily in correct, but has informal tones or unconventional phrasing, that should be addressed by a copy editor. There are also a large number of places with incorrect grammar, unexpected paragraph breaks, and inconsistencies between sections. I have pointed out some of these, but certainly not all of them. In addition, the use of abbreviations is inconsistent.

Author Response

Overall:
This is a very interesting and potentially important study examining the toxicity of quail egg oil. Since quail egg oil may have a large number of beneficial properties, making it suitable for a variety of uses, identifying any potential toxic side effects greatly enhances understanding of when it is appropriate to use it medically and in other commercial applications. I like and appreciate the historical and ethnographic perspective in the introduction, explaining traditional and current practices using egg oils. Combining this with the extensive scientific testing described here is a great strategy for developing new approaches to maintaining and improving health.

I commend the authors for writing a manuscript so well in a language that is not their native one. I would not do nearly this well if trying to write in a language other than English. However, there are several places where phrasing is awkward… not necessarily in correct, but has informal tones or unconventional phrasing, that should be addressed by a copy editor. There are also a large number of places with incorrect grammar, unexpected paragraph breaks, and inconsistencies between sections. I have pointed out some of these, but certainly not all of them. In addition, the use of abbreviations is inconsistent.

Specific Comments:
Line 20-21: I am a little confused about how many tests were used. I think bovine corneal opacity and permeability is one test, but with the comma in the middle, it looks like two. Is Histology ex vivo something separate? I would assume so, but again the punctuation is confusing.
    Thank you so much for your observation. Bovine corneal opacity and permeability tests with the histology of the bovine eye are combined or performed separately.  In this article, the bovine eye was inoculated, and histology of the bovine eye was directly used alone because it gives the best results. This was fully amended in the methodology and the whole write-up.
Line 26: does “both extracts” refer to heat-treated and NHN?
    Yes, it refers to the two extracts.
Line 33: what are the cardio-protective properties other than lower blood pressure? Or is that specifically a result of lower blood pressure
    Protecting the heart against oxidative stress and inflammatory insult
Line 35: just want to make sure that I understand… you are not proposing that its use in eye drops and 
cosmetics will prevent hypertension, but rather that these are all appropriate separate uses?
    Yes
Line 41 & 60: “active principle”???
    corrected
Line 47: “into” is a pretty informal way to say it… maybe “chose to” or “participate in”?
    corrected
Line 51: awkward phrasing “never been an issue to people using it”.
    Rephrased 
Line 62: instead of “minerals”, use “micronutrients” here
    Thank you, it has been corrected, please see line 69.
Line 66: I believe these are both important in clotting… this should be stated.
    It has been stated, please see line 73-74.
There are some organization issues in the introduction. For example, the paragraph starting on line 55 
should probably be combined with the paragraph on line 71. Maybe put all the ethnographic 
information together first, and then follow with content and scientific evidence for health/medical uses?
In general, it would be useful to combine paragraphs about very similar topics together, for example the 
paragraph starting on line 83 could very reasonably be combine with the one before it. Similarly the last 
2 paragraphs of the intro should be combined into one.
    Thank you for your observation, the introduction was re-arranged as suggested. Please see lines 47-77.
Line 80: Instead of “show symptoms” how about “cause obvious symptoms”
    Corrected. Please see line 83.
Line 86: Not clear what kinds of drugs you are talking about, or better efficacy than what. I think you 
mean something like “Many new drugs have been introduced to treat hypertension which may 
demonstrate better efficacy than existing hypertension drugs, but possess side effects.” Or maybe you 
mean better efficacy than quail egg oi? 
    Thank you, it has been corrected. Please see lines 89-91.
Lines 90-93: What do you think of: “In view of claims regarding the medicinal effect of quail eggs, this 
study was designed to quantify the efficacy of oil extracts from quail eggs at decreasing blood pressure
through analysis of serum antioxidant and hypertension markers. In addition, we assess the toxicity of 
quail egg oil extract using both in-vitro and in-vivo models.”
    Thank you. Corrected. Please see lines 94-98.
Lines 104-105: remove “are from”, “while” and “were from”
    Removed. See line 110.
Line 127: if possible, more details about what you mean by gently and by suitable cloth filter. A 
temperature or description of heating method, and the type of cloth used in the filter, are appropriate.
    Thank you. More information provided.please see lines 133-136.
Line 143: this is not a sentence.
    corrected
Line 150: I don’t understand what (group of 3 to 6 corneas by objectives test) means.
    Corrected. Please see lines 156-158.
Lines 147-165: this paragraph is grammatically very confusing. I don’t understand what was done.
Corrected. Please see Lines 158-167.
Line 167: what about moderate? Also, shouldn’t location and severity be considered separately? 
Line 196: what kinds of lesions?
    Corrected. Pathological leisons.
Should sections 2.11, 2.12, 2.13 be subsections (or just paragraphs within) section 2.10? Similarly, 2.14 
through 2.17 seem like subsections of 2.13.
Line 206: are these the same 30 rats discussed in 2.14?
No, they are different ones.
Line 222: I think this is the negative control?
2.20 should be a subsection of 2.19
    Amended.
Line 341: I don’t think this is a percentage. If it is, than so should the numbers on line 340 be.
    Corrected.
Fig. 2: is this showing corneal opacity or pupil dilation?
    Amended. It is showing the gross appearance of the cornea after treatment.
Lines 361-380: Is this all part of the figure legend? I don’t think it should contain all the “Fig.”s because these are not referring to figure numbers but rather parts of Fig. 3. Either way, there should be a resultssection about this figure that explains what these images mean… not just what tissues, but what is normal or what different types of abnormality are seen.
    Corrected, please see lines 374-390

Line382: QEYO should not be in parentheses
    Corrected.
Line 383-386: add “rats” before “showed”… maybe try “rats showed no sign of toxicity after 48 hours of monitoring with….” Similarly the next sentence needs a subject in the first part.
    Corrected, please see lines 412-416.
The last 2 sentences of sections 3.1 and of 3.2 are identical.
    Corrected please see lines 413-422.
Tables 2, 3 & 4 should indicate p-values for differences between the three treatments, or differences between each treatment and the control. If none of the differences are significant, this could go in the table description rather than the body of the table. Since the categories in the headers are all the same, they should be presented in the same way, preferably as they are in Table 3. I assume that in T. protein the T is for total. Not sure what D. is in D. bilirubin.
    Corrected, please see Tables 2,3&4
    In the liver, bilirubin is changed into a form that your body can get rid of. This is called conjugated bilirubin or direct bilirubin. This bilirubin travels from the liver into the small intestine. A very small amount passes into your kidneys and is excreted in your urine. This bilirubin also gives urine its distinctive yellow color. This test is often done to look for liver problems, such as hepatitis, or blockages, such as gallstones.

“Blood pressure” could all be one section. You don’t need separate sections for baseline, after L-NAME administration, and after treatment.
    corrected
Don’t the first and second paragraphs of section 3.8 say basically the same thing? 
corrected
Line 465-66: This does not agree with what is shown in the graph, which is that QEYOGN 300 mg/kg showed the greatest reduction, followed by Nifedipine. QEYONH 300 mg/kg showed the smallest reduction compared to control. Not sure if the text is wrong or the graph is mislabeled?
    corrected, please see lines 500-502.
Figure 6 does appear to have lines mislabeled, as purple and orange show the same treatment. Colors should match the same treatment in Fig. 5 and 6.
    corrected
Figure 8: formatting issues
    corrected to figure 7
Sec. 3.12 goes before Table 8.
    Amended
Line 518-519: This is not a sentence. It also doesn’t make sense … which extract of QEYO, as you already said that QEYONH has some differences.
    deleted
Line 550: QEYO should probably be the end of this sentence. In fact, L-NAME induced hypertension is a completely different topic and might even be appropriate as a new paragraph. 
    corrected
Line 566: sodium concentration is lower compared to what? Also, the effect of calcium levels on the heart has not changed recently. 
    Corrected
Line 573: what do you mean by “tone”?
corrected
Line 590-594: this sentence is confusing. I’m not sure I understand what your claim here is.
    corrected
Lines 595-596: are these other agents related to QEYO?
    deleted
Line 626: what do the early reports indicate?
    Included 662-670.
Line 638: could decrease what? Or decrease in response to what? This is confusing.
    Amended
Line 643: You don’t really talk a lot about kidney function. That should be explained more in the results. and discussion if it is to remain in the conclusion. The conclusion should also mention the lack of toxic effects. 
    amended
I don’t see where reference #54 is cited
    corrected
The power of this paper is that you both show no toxicity AND beneficial effects together

Reviewer 2 Report

Comments and Suggestions for Authors

This manuscript clearly describes the methods, which are appropriate, and the results.  A quick literature search has located: Rehault-Godbert et al (2019) The Golden Egg: Nutritional Value, Bioactivities and Emerging Benefits for Human Health, which appears to have very good background information and might be worthy of citation.

There is some attempt to discuss the potential mechanism of the antihypertensive effect, but I believe that this could go further.  Egg yolk is known to contain proline and arginine, both of which may decrease blood pressure.  Would these amino acids survive the extraction process?  Could they be responsible for the observed effects?

Add Comments: The manuscript had minor typographical / grammatical errors that can easily be corrected. The work is good in that it demonstrates a potentially beneficial effect of quail egg extract on blood pressure and recognises the importance of considering potential toxicity before heralding clinical potential.

The cited references are good, but I indicated a manuscript that I identified in a superficial literature search which could add something to the manuscript in terms of the composition of the quail egg extract. Importantly, I think that the authors could enhance their discussion of the possible mechanism of action of the extract by considering the amino acid content of egg yolk, and the possible effects of those amino acids on blood pressure.

Author Response

Reviewer 1:

Comments and Suggestions for Authors

This manuscript clearly describes the methods, which are appropriate, and the results.  A quick literature search has located: Rehault-Godbert et al (2019) The Golden Egg: Nutritional Value, Bioactivities and Emerging Benefits for Human Health, which appears to have very good background information and might be worthy of citation.

  • Thank you so much for your comment. The reference was found very useful and have been updated accordingly.

There is some attempt to discuss the potential mechanism of the antihypertensive effect, but I believe that this could go further.  Egg yolk is known to contain proline and arginine, both of which may decrease blood pressure.  Would these amino acids survive the extraction process?  Could they be responsible for the observed effects?

  • Thank you so much for you comment. The proposed mechanisms were fully elucidated in the discussion part. Please see lines 660-668.

Add Comments: The manuscript had minor typographical / grammatical errors that can easily be corrected. The work is good in that it demonstrates a potentially beneficial effect of quail egg extract on blood pressure and recognises the importance of considering potential toxicity before heralding clinical potential.

  • Thank you so much for your comments. The typographical/grammatical errors are corrected.

The cited references are good, but I indicated a manuscript that I identified in a superficial literature search which could add something to the manuscript in terms of the composition of the quail egg extract. Importantly, I think that the authors could enhance their discussion of the possible mechanism of action of the extract by considering the amino acid content of egg yolk, and the possible effects of those amino acids on blood pressure.

  • Thank you for your comment. The article reference has been utilized both in the introduction as well as in the discussion.

Round 2

Reviewer 1 Report

Comments and Suggestions for Authors

Thank you for making the changes to the manuscript. I think it is much more clear and makes your important points more strongly. This is an important topic and I am looking forward to the manuscript being published.

Comments on the Quality of English Language

Please check through one more time for typos, as I did still see a few. In addition, there were a few incomplete sentences, for example lines 336-37 and the first sentence of section 3. I believe I saw another one, but didn't write it down and now I don't see it, so check through that all sentences have verbs. Please also check the figure legends for legibility. I did notice that parts were cut off for Figs. 5 & 6.

Author Response

Response to reviewer:

Thank you for your comments on the document, it improved the quality of the document. 

Please check through one more time for typos, as I did still see a few.

Response: I have read through the entire document and corrected the typos.

In addition, there were a few incomplete sentences, for example lines 336-37 and the first sentence of section 3. I believe I saw another one, but didn't write it down and now I don't see it, so check through that all sentences have verbs.

Response: I have fixed the grammatical issues highlighted above and made necessary corrections throughout the document

Please also check the figure legends for legibility. I did notice that parts were cut off for Figs. 5 & 6.

Response: I have made the necessary corrections to the legends in Fig 5 and 6.